# Self-Reported Psychosomatic Complaints and Conduct Problems in Swedish Adolescents

**DOI:** 10.3390/children9070963

**Published:** 2022-06-27

**Authors:** Samantha J. Brooks, Olga E. Titova, Emma L. Ashworth, Simon B. A. Bylund, Inna Feldman, Helgi B. Schiöth

**Affiliations:** 1Functional Pharmacology and Neuroscience, Department of Surgical Sciences, Uppsala University, 751 24 Uppsala, Sweden; helgi.schioth@neuro.uu.se; 2Faculty of Health, School of Psychology, Liverpool John Moores University, Liverpool SE3 3AF, UK; e.l.ashworth@ljmu.ac.uk; 3Neuroscience Research Laboratory (NeuRL), Department of Psychology, School of Human and Community Development, University of the Witwatersrand, Johannesburg 2000, South Africa; 4Unit of Medical Epidemiology, Department of Surgical Sciences, Uppsala University, 751 24 Uppsala, Sweden; olga.titova@surgsci.uu.se; 5Uppsala County Council, 751 25 Uppsala, Sweden; simon_bylund@hotmail.com; 6Department of Public Health and Caring Science, Uppsala University, 751 85 Uppsala, Sweden; inna.feldman@pubcare.uu.se

**Keywords:** conduct disorder, psychosomatic complaints, adolescence, multivariate analysis

## Abstract

Physical conditions in children and adolescents are often under reported during mainstream school years and may underlie mental health disorders. Additionally, comparisons between younger and older schoolchildren may shed light on developmental differences regarding the way in which physical conditions translate into conduct problems. The aim of the current study was to examine the incidence of psychosomatic complaints (PSC) in young and older adolescent boys and girls who also report conduct problems. A total of 3132 Swedish adolescents (age range 15–18 years, 47% boys) completed the Uppsala Life and Health Cross-Sectional Survey (LHS) at school. The LHS question scores were categorised by two researchers who independently identified questions that aligned with DSM-5 conduct disorder (CD) criteria and PSC. MANOVA assessed the effects of PSC, age, and gender on scores that aligned with the DSM criteria for CD. The main effects of gender, age, and PSC on the conduct problem scores were observed. Adolescents with higher PSC scores had higher conduct problem scores. Boys had higher serious violation of rules scores than girls, particularly older boys with higher PSC scores. Psychosomatic complaints could be a useful objective identifier for children and adolescents at risk of developing conduct disorders. This may be especially relevant when a reliance on a child’s self-reporting of their behavior may not help to prevent a long-term disturbance to their quality of life.

## 1. Introduction

Mental health difficulties in children and adolescents are increasing, with estimated global prevalence rates between 10–20% [1], although the maximum range could be as high as 75% [2]. Detecting mental health difficulties in children and adolescents can be challenging, with a reliance on subjective reporting that may be sparse, inaccurate, may differ from parental reports, and may not adequately reflect their underlying mental health issues [3,4]. It may be easier for children and adolescents to report physical issues than to self-report on mental health [5], yet the extent to which physical complaints are associated with behavioral and mental health difficulties in children and adolescents is not clear. In addition, the moderating and mediating effects of emotional problems can alter the relationship between mental health difficulties and behavioral issues [6]. For example, deficits in the neural processes of affect regulation may be mediated by childhood trauma and maltreatment [7], or prenatal maternal somatic diseases [8], which may cause a myriad of mental health difficulties that are antecedents of behavioral issues. Furthermore, the perception of emotional and behavioral autonomy provided by parents could exacerbate a child’s internal distress or deviant behavior, respectively [9]. However, the moderating effects of emotional problems that are independent of potential underlying mental health difficulties, such as perceived socioeconomic status [10] and nutrition [11], for example, may also lead to behavioral problems in adolescents.

The diagnosis of a conduct disorder (CD) is an objective measure of behavioral difficulties that may coincide with mental health difficulties in young people [2], with 3–4% of adolescent boys and 1–2% of adolescent girls receiving a CD diagnosis [3]. Yet, the incidence of CD is sufficient but not necessary for detecting underlying mental health issues in the young, as there are still a high number of children and adolescents with mental health issues that do not have a formal diagnosis of CD. While many young children present with mild behavioral difficulties during development, CD is only diagnosed when a child’s behavior becomes extreme: outside the norm for the age and level of development [12,13,14]. A diagnosis of CD occurs, according to the fifth Edition of the Diagnostic and Statistical Manual for mental disorders (DSM-5) [15], if a child or adolescent meets 3 out of 15 criteria for disordered behavior over a 12-month period across four categories: aggression to people/animals, deceit and theft, destruction of property, and a serious violation of rules (e.g., school truancy or prolonged absences from home). Furthermore, the age of onset and limited prosocial emotions (LPE, or callous/unemotional traits) are CD specifications recently added to DSM-5 that further determine severity [16,17]. Given the high incidence of adolescent mental health difficulties, and that CD is only diagnosed in a small proportion of extreme cases, the use of other measures in typical school children may better identify an early risk for mental health and conduct difficulties [18].

One such measure could be the self-reporting of psychosomatic complaints (PSC), including headaches/migraines, stomachaches, backaches, fatigue, anxiety, and depression, as some research shows that PSCs precede behavioral problems, including disengagement with school or home rules, and oppositional/defiant behaviors [19,20]. For example, in a recent Swedish multivariate study across 2000 schools, 60,000 young adolescents’ likelihood of being bullied and their mental health complaints were examined [21]. The study found that being the perpetrator of bullying predicted a higher incidence of PSCs and behavioral problems. In addition, a large longitudinal study of adolescents across a 23-year period found that self-reported PSCs translated into the deterioration—over the study period—of cognitive-behavioral health [22]. In terms of specific PSCs, a study of 5730 adolescents reported that headaches were the most common physical complaint and were significantly correlated with the instigation of bullying and disruptive behavior at school [23]. However, caution must be taken when considering the causal relationship between PSCs and deviant behaviour such as bullying or other CD subtypes, because PSCs could also be the result of emotional trauma, affect dysregulation, or some other cause (e.g., learned behaviour such as *modelling*).

In terms of the age of onset of PSCs and behavioral problems, considering the differences between younger versus older adolescents could be informative, given some key non-linear neurodevelopmental milestones in these two groups. For example, significant neural changes in frontostriatal circuitry underlying affect and impulse regulation are well known [24], suggesting that younger adolescents may be more prone to PSCs and behavioural problems (due to the inefficient self-regulation) than older adolescents. Conversely, older adolescents, particularly girls, appear to demonstrate diminishing mental health in some cohorts [25]. Moreover, in a study of younger children it was found that victims of bullying were at a higher risk of PSCs such as sleep problems and feeling tense, tired, or dizzy, as well as later behavioral problems [26]. However, while bullying may be associated with certain domains of CD, studies into bullying do not directly examine the link between PSC and conduct problems. Further research may highlight a specific link between (a) the presence of PSCs, (b) minor behavioral issues that if left undetected may transform into (c) formal CD, and the underlying mental health difficulties that may contribute to all three. For example, in another recent study examining the same Swedish cohort as the present study, it was demonstrated that underweight vs. overweight boys and girls had higher PSCs (e.g., headache and pain in the hips) and that these complaints significantly contributed to mental health difficulties (e.g., anxiety, depression, and generally ‘feeling low’), although this study did not consider DSM5 CD-related behavioral issues [27]. Another Swedish study from a different, large cohort, found that subjective health complaints in adolescents were predictive of higher stress levels and mental health issues [28]. Thus, these studies indicate that early subjective reports of psychological and somatic complaints in young children may be better indicators of underlying distress than behavioral issues that may eventually become a formal diagnosis of DSM5 CD.

To the authors’ knowledge, no large sample analysis has yet examined whether self-reported adolescent behavioral problems from a government-commissioned survey in otherwise healthy mainstream school students are associated with a higher incidence of self-reported PSC. Therefore, the aim of the present study was to explore this in a Swedish sample, utilising the Uppsala Life and Health Young Cross Sectional Survey (LHS) commissioned by Uppsala County Council, Sweden, between 2005 and 2011 and conducted in a subsample of school-age children aged 15–18 years. Survey questions were linked to DSM-5 CD criteria and PSC by two independent researchers who were unaware of the other’s categorisations, to clarify the selection of specific questions from the government-commissioned survey. It was hypothesized that: (a) high scores on self-reported PSC questions would be related to high scores on self-reported behavioral issues, (b) gender differences would be observed in the scores of PSC and behavioral issues, and (c) younger versus older adolescents would have significantly different self-report scores of PSC and behavioral issues.

## 2. Materials and Methods

### 2.1. Design and Participants

An initial cross-sectional sample of 39,399 adolescents aged 12–19 years (the subsequent study sample consisted of 15–18-year-olds, see below) was invited to participate anonymously and voluntarily in the Life and Health Young Cross-sectional Survey (LHS), commissioned by the Uppsala County Council, Sweden, to be conducted in schools in separate cohorts in 2005, 2007, 2009, and 2011. Details of the full LHS questions are available on request. All respondents were asked to complete the survey only once, in one of these years, during school hours, and no identifiable personal data (e.g., date of birth, name, or exact home location) was collected by the school researchers. Consent from parents and assent from pupils was collected, and both pupils and parents were made aware during introduction to the survey and the study itself that they could withdraw at both their participation and data at any time, without their rights being affected. In 2005 and 2007, data on age, household structure, and drug use were not available, and so only years 2009 and 2011 (which collected such data) were included in the current study. Thus, a total of 9667 adolescents were initially eligible for the present analysis. From this total, a further 99 adolescents were excluded for not having all data for age, gender, and PSC, leaving a total of 9568. Secondary exclusions of 6436 participants were due to at least one missing answer on questions contributing to the total score of the conduct problems that was determined by two independent researchers (see above) to be related to total DSM-5 CD criteria and the sub-categories of CD (described below), leaving a total of 3132 for the main analyses.

The final analysis consisted of adolescents aged between 15–18 years with no missing values. A rudimentary attrition or churn rate analysis (e.g., taking the number of respondents with missing values = 36,267, and dividing by the number of original participants = 39,399) suggested a 92% attrition rate in this survey study of Swedish adolescents. It must be noted that the analysis of only complete cases is somewhat problematic as it violates the intention to treat principle, such that representative conclusions for this population are significantly reduced [29]. The flowchart of the study population exclusions is shown in Figure 1. Data analysis of this cohort was approved by the Ethical Committee of Uppsala (EPN) with permission from Uppsala County Council. The dataset used for this study is available on request.

### 2.2. Measures

Demographic data. Age and gender were examined as recorded in the LHS. Age was dichotomized according to mean split (above versus below the mean (age 16) for older (17–18 years) versus younger (15–16 years) adolescents, respectively), and gender was coded 0 for males and 1 for females.

Blind independent rating. Two independent, experienced researchers (a medical doctor/researcher, and a cognitive neuroscientist) independently examined the entire LHS set of questions (SBAB, SJB). Each researcher was blind to the other’s choices when selecting self-reported conduct problem questions corresponding to DSM5 CD subcategories (aggression to people/animals, deceit and theft, destruction of property, and serious violation of rules [e.g., school truancy or prolonged absences from home]), and to psychosomatic complaints (PSC). Any discrepancies in choices between the two researchers were discussed, and if inclusion/exclusion of questions for CD or PSC categories could not be agreed upon, a third researcher with publications in adolescent mental health research (HBS) was consulted to aid the final decision on the few occasions this occurred.

Self-reported conduct problems. Conduct problem scores were created by summing scores for questions with Likert-scale questions that the two independent researchers categorised according to the 4 DSM-5 CD criteria, as summarised in Table 1. Combining Likert scale scores for several questions (per CD category in this case) is a method for creating continuous, quantitative data, for which the mean central tendency can be used, and is justified by the Central Limit Theorem for using parametric techniques such as ANOVA (Norman, 2010). Based on the summed Likert scale scores, higher conduct problem scores corresponded to higher levels of behaviors associated with DSM5 CD subcategories. Details of each of the included LHS questions (with reference to the question numbers) and their scoring (no scores were reversed for the conduct disorder questions) are given in Appendix A.

DSM-5 CD criterion: Aggression to people or animals. 11 questions were selected from the total LHS to closely represent this criterion, with a cumulative minimum score of 11 and a maximum score of 49. No scores were reversed.

DSM-5 CD criterion: Destruction of property. 2 questions from the LHS were closely matched to this criterion, with a cumulative minimum score of 2 and a maximum of 10. No scores were reversed.

DSM-5 CD criterion: Deceitfulness or theft. 9 questions from the LHS closely matched this criterion, with a cumulative minimum score of 9 and maximum of 45. No scores were reversed.

DSM-5 CD criterion: Serious violations of rules. 10 questions from the LHS closely matched this criterion, with a cumulative minimum score of 10 and a maximum of 41 No scores were reversed.

Total conduct problems score. All 32 questions from the categories above were summed to create a cumulative LHS minimum score of 32 and a maximum score of 145 (according to the Likert Scale scores for each question). Percentages for total LHS scores and the 4 categories (e.g., according to the highest possible score) were calculated for contrast homogeneity.

Assessment of self-reported psychosomatic complaints (PSC). The same two independent researchers (SB and SBAB) examined all the LHS questions and categorised them according to either psychological complaints (e.g., anxiety or feeling low), or somatic complaints (e.g., headache or hip ache). A total of 12 LHS questions were related to psychological and 17 were related to somatic complaints. Percentages for PSC total and the two subscale scores for psychological and somatic (e.g., according to the highest possible score for total Likert Scale scores) were calculated for contrast homogeneity. Three levels of total PSC—low, medium, and high—were calculated as tertile thresholds of the percentage scores (one score for psychological complaints was reversed, see below). See Appendix A for details of each of the included questions and their scoring.

Psychological complaints. The 12 psychological LHS questions have a cumulative minimum score of 12 and a maximum score of 52 and are related to: stress, nervousness, anxiety/worry, depression, happiness, medication for anxiety/depression/sleep disorder, reading/writing difficulties, neuropsychiatric disorder (e.g., ADHD), worries about sleep, occurrence of nightmares, and how bright the future looks. One question (B5_13R) asking whether the respondent was happy was reverse-scored. No other scores were reversed.

Somatic complaints. The 17 somatic (physical) LHS questions have a cumulative minimum score of 17 and a maximum score of 70 and are related to: how the person feels; incidence of headache, migraine, or stomach ache; ringing in the ears/tinnitus; tiredness; pain in the neck and shoulders; pain in the back and hips; pain in the hands, knees, legs, or feet; quality of dental health; whether prescription or non-prescription drugs are taken for headaches or other pain; hearing loss; visual impairment that cannot be correct with glasses; physical disability; difficulty sleeping; and whether the person is often tired. No scores were reversed.

Total PSC score. All 29 LHS questions above were summed to create a cumulative minimum score of 29 and a maximum score of 122, which allowed for the calculation of individual total percentage PSC scores.

### 2.3. Statistical Analyses

SPSS version 27.0 (IBM Corp, 2020, Armonk, NY, USA) was used. Descriptive statistics of the demographic data were conducted with *t*-tests for mean differences and chi-squared for frequencies, where applicable. Bonferroni correction was applied to control for multiple comparisons, and parametric assumptions (normal distribution, homogeneity of variance, sphericity, and collinearity) were checked, and data were corrected if these assumptions were violated. A *p*-value of equal to or less than 0.05 denoted significant effects.

Analysis of variance (ANOVA) of total conduct problems score (*n* = 3132). A 3-way independent measures design, incorporating the between-subject factors of age (dichotomised above and below the mean age of 16 years), gender (boys and girls), and PSC severity score (tertiles for low, medium, and high), was utilised to examine the main effects and interactions associated with variance in the total LHS behavioral problem score. Post-hoc independent *t*-tests were used to determine the direction and significance of any main effects on measures of total conduct problems score, or interactions between age (younger vs. older), gender (male or female) or PSC (low, mid, or high). See Figure 2.

We planned to run multivariate analysis of variance (MANOVA) of the 4 categories of conduct problems (*n* = 3132) as a 3-way independent measures design, incorporating the between-subject factors age (dichotomised above and below the mean age of 16 years), gender (boys and girls), and PSC severity score (tertiles for low, medium, and high), to examine the main effects and interactions associated with a score variance in the four conduct problem categories that were linked to DSM-5 CD, namely: (a) aggression to people/animals, (b) deceit/theft, (c) destruction of property, and (d) serious violations of rules. However, it was found that the subscales for CD were highly correlated (see Table 2), and so separate ANOVAs were calculated for each of the 4 subscales. Again, post-hoc *t*-tests were used to determine the direction and level of significance of any differences between the main effects of age on individual conduct disorder scores or the interactions. In all statistical analyses, a *p*-value of 0.05 with Bonferroni correction applied (*p* ≤ 0.02) was deemed significant.

## 3. Results

### 3.1. Demographics

A total of 1460 boys and 1672 girls contributed to the main ANOVA (examining the effects on total conduct disorder score) and to the four sub-ANOVA analyses (examining the four sub-scales of conduct disorder) in a total sample of 3132 adolescents after exclusions. According to a mean age split at 16 years of age, 1533 were ‘younger adolescents’ (15–16 years of age) and 1599 were ‘older adolescents’ (17–18 years of age). A total of 1321 adolescents were rated as having low-, 868 with medium-, and 943 with high-PSC severity. No significant difference in age was detected in the sample of boys and girls. Adolescent girls reported more high PSCs than boys, specifically, they reported a higher severity of headaches, stomach aches, and tiredness, as well as higher levels of anxiety and depression (see Table 3).

### 3.2. Statistical Analyses

#### 3.2.1. Analysis of Variance (ANOVA) of Total Conduct Problems Score (*n* = 3132)

The effects of gender (boys and girls), age (younger and older adolescent), and PSC severity (low, medium, and high) on the total conduct problems score were examined. Significant main effects of gender (F[1, 3000] = 124.220, *p* < 0.001, η^2^ = 0.038), age (F[1, 3000] = 21.664, *p* < 0.001, η^2^ = 0.007), and PSC severity (F[2,3000] = 77.342, *p* < 0.001, η^2^ = 0.046) were observed. No significant interactions were observed. Post-hoc *t*-tests confirmed that boys had significantly higher total conduct problems scores than girls (t[2898.579] = 8.114, *p* < 0.001), older adolescents had higher conduct problems scores than younger adolescents (t[3459] = 4.699, *p* < 0.001), and all adolescents regardless of age or gender with high PSC severity had higher conduct problem scores relative to those with medium (t[2053.938] = 7.000, *p* < 0.001) or low (t[2203.242] = 9.333, *p* < 0.001) PSC severity. Those with medium PSC severity had higher conduct problem scores than those with low severity (t[2038] = 2.630, *p* < 0.009). See Figure 2 and Table 4 for an illustration of the scores across gender for conduct problem score differences.

#### 3.2.2. Separate ANOVAs for the Four Categories of Conduct Problems (*n* = 3132)

The effects of gender (boys and girls), age (younger and older adolescents), and PSC severity (low, medium, and high) on the four categories of conduct problems (aggression to people/animals (APA), deceit/theft (DT), destruction to property (DP), and serious violation of rules (SVR)) were examined separately, given that each of the four dependent variables were correlated (see Table 5).

##### Aggression to People/Animals (APA)

A main effect of gender on APA percentage scores was observed (F[1, 3120] = 119.072, *p* < 0.001, η^2^ = 0.037), and post-hoc *t*-tests confirmed that boys had higher scores compared to girls (t[2215.523] = 8.302, *p* < 0.001). A main effect of PSC severity on APA percentage was observed (F[2, 3120] = 38.052, *p* < 0.001, η^2^ = 0.024), with post-hoc *t*-tests revealing that high PSC severity had higher APA scores than low PSC severity (t[2143.124] = 4.807, *p* < 0.001) and medium-severity (t[1966.526] = 4.135, *p* < 0.001), but there was no difference in APA scores between low and medium PSC severity. No other significant main effects or interactions were observed.

##### Deceit/Theft (DT)

A main effect of gender on DT percentage scores was observed (F[1, 3120] = 51.024, *p* < 0.001, η^2^ = 0.016), and post-hoc *t*-tests confirmed that boys had higher scores compared to girls (t[2469.503] = 4.359, *p* < 0.001). A main effect of PSC severity on DT percentage was observed (F[2, 3120] = 42.778, *p* < 0.001, η^2^ = 0.027), with post-hoc *t*-tests revealing that medium-severity PSC had higher DT scores than low-severity (t[2012] = 1.663, *p* = 0.048), and that high-severity PSC had higher DT scores than low-severity (t[2159.228] = 7.029, *p* < 0.001) and medium-severity PSC (t[2001.720] = 5.648, *p* < 0.001). No other significant main effects or interactions were observed.

##### Destruction to Property (DP)

A main effect of gender on DP percentage scores was observed (F[1, 3120] = 54.918, *p* < 0.001, η^2^ = 0.017), and post-hoc *t*-tests confirmed that boys had higher scores compared to girls (t[2731.444] = 5.359, *p* < 0.001). A main effect of PSC severity on DP percentage was observed (F[2, 3120] = 25.031, *p* < 0.001, η^2^ = 0.016), with post-hoc *t*-tests revealing that medium-severity PSC had higher DT scores than low-severity (t[1926.459] = 2.213, *p* = 0.014), and that high-severity PSC had higher DP scores than low-severity (t[2158.842] = 5.024, *p* < 0.001) and medium-severity PSC (t[2024.955] = 52.737, *p* = 0.003). No other significant main effects or interactions were observed.

##### Serious Violation of Rules (SVR)

A main effect of gender on SVR percentage scores was observed (F[1, 3120] = 79.780, *p* < 0.001, η^2^ = 0.025), and post-hoc *t*-tests confirmed that boys had higher scores compared to girls (t[3130] = 4.610, *p* < 0.001). A main effect of age on SVR scores was observed (F[1, 3120] = 145.985, *p* < 0.001, η^2^ = 0.045), with post-hoc *t*-tests revealing that older children had higher SVR scores than younger children (t[3129.327] = 12.584, *p* < 0.001). A main effect of PSC severity on the SVR scores was also observed (F[2, 3120] = 82.268,*p* < 0.001, η^2^ = 0.050), with post-hoc *t*-tests revealing that children with medium-PSC scores had higher SVR scores than those with low PSC scores (t[1870.362] = 3.273, *p* < 0.001), and that those with high PSC scores had higher SVR scores than those with medium PSC (t[2015.550] = 7.751, *p* < 0.001) and low PSC scores (t[2138.231] = 11.642, *p* < 0.001). A trend interaction (*p* = 0.057) was observed between gender and PSC severity on SVR scores (F[2, 3120] = 2.867, *p* = 0.057, η^2^ = 0.002), in that older boys with the highest PSC severity scored the highest for SVR (M = 49.84, SD = 10.848), whereas the lowest SVR score was for younger girls with the lowest PSC severity (M = 37.46, SD = 7.030). the interaction trend data are available on request.

## 4. Discussion

To the best of our knowledge, this is the first study to show that in an otherwise healthy population of Swedish adolescents attending mainstream school, high levels of psychosomatic complaints (PSCs) associate with higher levels of total conduct problems. This association was amplified for boys who scored highly in the serious rule violations category, especially those with higher PSC severity scores displaying higher levels of serious rule violations. In addition, older adolescents (17–18 years of age) reported higher levels of PSCs and conduct problems than younger adolescents (15–16 years of age), and older boys reported significantly higher levels of conduct problems than younger girls.

The finding that adolescents in this sample, particularly boys aged 17–18 years, exhibit higher levels of non-aggressive executive dysfunction in the form of serious rule violations, associated with an increasing PSC, provides some clues as to the potential underlying factors associated with conduct problems. These findings suggest that increased somatic complaints early in life may contribute to altered neural development associated with top-down emotion processing and self-regulation, leading to a propensity towards conduct problems [24].

Conversely, altered neural development may contribute to low prosocial emotion/callous unemotional traits that increase the risk of having conduct problems [6,7]. In addition, effective emotion regulation and adaptive decision-making relies on somatic processing (e.g., the somatic marker hypothesis), or a ‘gut feeling’ [30], which is necessary for cooler executive functions, including empathy, understanding the future consequences of actions, and the learning of beneficial decisions and adherence to societal rules [31]. As such, gauging and addressing common adolescent PSCs, including frequent headaches, stomach aches, tiredness, anxiety, and depression, may reduce the risk of conduct problems—particularly in boys—and the future development of adult mental health disorders, such as antisocial personality disorder (common in males), anxiety and depression (common in females), and other psychopathologies [32]. Together with the previous literature, our findings suggest that conduct problems may have at least part of their origin in being mentally and/or physically afflicted by PSCs during childhood and adolescence. That said, other mediating and moderating influences on mental and behavioural disorder (e.g., childhood trauma, fetal alcohol syndrome, maternal somatic states, socioeconomic status, and nutrition) should also be considered [7,8,9,10,11].

Specific PSCs, namely stomachache, tiredness, and problems related to anxiety were associated with self-reported conduct problems in the present study. However, while this association was strongest for boys, it was girls reporting higher PSC scores. This may be explained by the measure of mental health difficulties used (i.e., conduct problems). It is possible that there is a gender difference in the presentation of mental health difficulties or in coping strategies for PSCs between boys and girls. For instance, while PSCs were associated with conduct problems in the current study, previous research has also found that childhood somatic complaints can predict generalised anxiety and depression in adolescents [33]. As boys are more likely to experience externalising difficulties (such as the behaviour problems explored here), and girls are more likely to present with internalising difficulties (such as anxiety and depression) [34], this may explain the stronger association between PSCs and conduct problems in boys in the present study. It may be that PSCs contribute equally to mental health difficulties, but the way these difficulties manifest themselves differs by gender. If this is the case, had an internalising measure of mental health difficulties been used, the gender effect might have been reversed. Future research could seek to further explore this possibility, investigating gender differences in the relationship between PSCs and types of mental health difficulties.

Finally, a striking point about the findings of this study is that the sample did not comprise adolescents with a formal DSM-5 CD diagnosis but those who were otherwise healthy and attending mainstream school. Thus, the relationships observed in the present study may be even stronger in individuals with a diagnosis of CD. Furthermore, the results from the present study highlight the relatively higher levels of conduct problems and PSC reported by adolescents in a general population sample, which may remain untreated if undiagnosed. This is problematic for individuals and society, given that adolescents with recurring behavioral problems are 60% more likely to develop serious mental health disorders, such as anti-social personality disorder, in adulthood [18]. As such, these data provide a further motivation to reverse the present trend wherein CD remains the least understood and studied psychiatric disorder [18].

### Strengths and Limitations

One of the main strengths of this study is the large sample size, and the calculation of scores from a large range of questions that were confidentially and anonymously answered by adolescents at school. In addition, two researchers (SJB and SBAB) independently linked the LHS questions to DSM-5 CD criteria. However, some limitations include the reliance on adolescent self-reports that were not formally related to DSM-5 CD criteria, which may not be as accurate as behavioral observations or the use of validated CD measures. In addition, this study includes quite old data (from 2011); however, we argue that the links between psychosomatic complaints and problem behaviours in children are less culturally-bound than formal DSM-5 criteria, and so are less susceptible to changes over time. Furthermore, this was a cross-sectional as opposed to a longitudinal study, and so conclusions regarding the differences in age and the causal relationships must be considered cautiously. In addition, there was a high drop-out rate, in that 36,267 of the original 39,399 sample had missing scores and were excluded from the current analysis, thus violating the intention to treat principle. Our drop-out rate highlights the importance of ensuring full data collection (e.g., by adding continuation limits to online surveys, such that participants must respond to each question before moving on to the next). Moreover, it is imperative to collect more recent data, as the last data in this sample were collected in 2011. Finally, while efforts were taken by the authors to closely match the LHS questions to DSM-5 CD criteria, the survey was not a validated measure of CD.

## 5. Conclusions

The main implications of this study are that if left undetected, psychosomatic complaints in younger children, especially boys, may later develop into serious rule violations and other conduct problems. This suggests serious consequences for socio-economic engagement, especially during school, continuing education, and employment prospects, where adhering to rules is essential for progress. As such, the present study aimed to better understand what might limit the performance of school-attending adolescents who may develop behavioural and mental ill-health later on. We considered the relationship between self-reported PSC and behavioral problems, to determine whether PSCs are a valuable gauge for early interventions to prevent the development of serious mental disorders into adulthood. If it is possible to link PSCs to potential future conduct problems, these can be used for earlier detection and intervention, to enhance the future well-being of children. The findings we present suggest that high levels of PSCs may be associated with higher levels of behavioural problems, and each of the four conduct problems subtypes linked to DSM-5 CD criteria, but most specifically for serious rule violation. This association is amplified for older boys. Neural dysfunction models of executive dysfunction may provide clues as to how early life PSCs may influence the brain development associated with decision-making, emotional regulation, and empathy. Children aged 15–17 (the age of the present cohort) undergo major developmental changes in the body and brain, and so the early detection of PSCs may have a huge impact on their later brain function and behaviour. An inability to effectively utilise somatic cues which are the basis of empathy and decision-making (e.g., due to a higher incidence of PSCs that may obscure the ability to attend effectively to somatic cues) may increase the prevalence of low prosocial emotion, rule violation, anxiety, and depression. Furthermore, when combining the findings from the present study with previous research, it is possible that PSCs contribute to mental health difficulties in both genders, but that coping mechanisms or presentation of symptoms differ between boys and girls. These data suggest that the early screening of children and adolescents at school for psychosomatic complaints may help to identify a need for intervention that could prevent the future development of behavioural problems and adult psychiatric conditions.

## Figures and Tables

**Figure 1 children-09-00963-f001:**
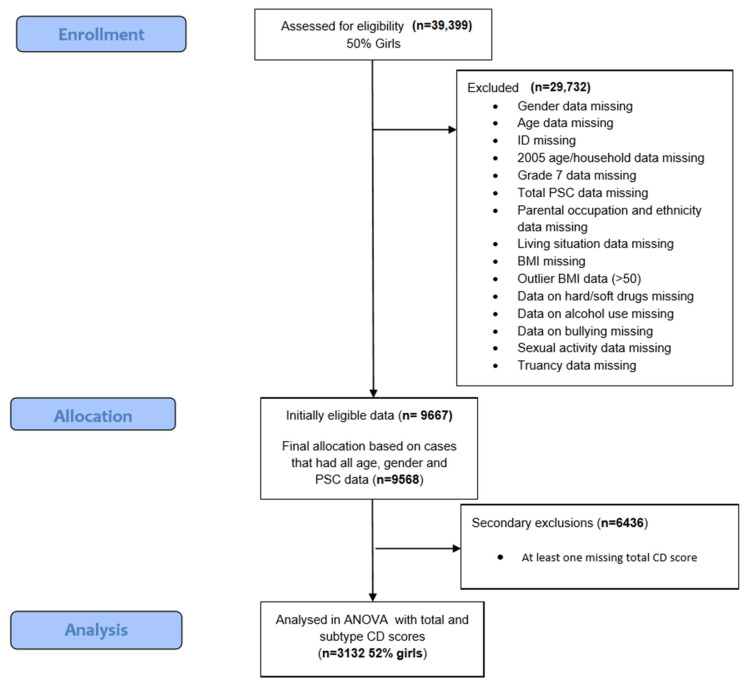
PRISMA diagram to describe participant enrollment, allocation and data analysis.

**Figure 2 children-09-00963-f002:**
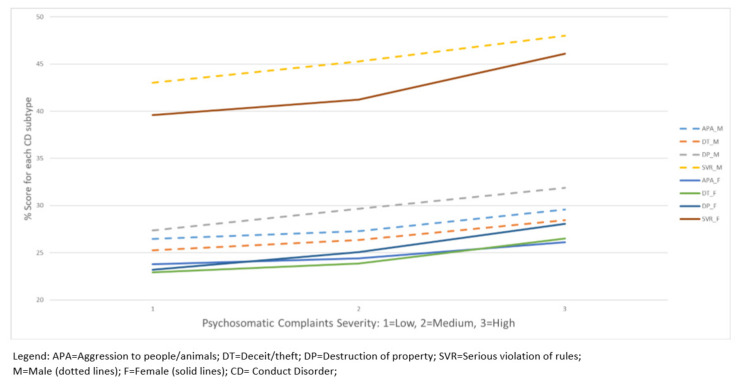
Behavioral Disorders scores in relation to levels of psychosomatic complaints in Sweden adolescents aged 15–18 years old.

**Table 1 children-09-00963-t001:** Diagnostic and Statistical Manual Version 5 (DSM-5) 15 Criteria for Conduct Disorder (CD) and the Sub-categories.

Category 1: Aggression: people or animals (APA)	
Item No.	Description
1	Often bullies, threatens or intimidates others
2	Often initiates physical fights
3	Has used a weapon that can cause serious physical harm to others (for example, a bat, a brick, broken bottle, knife or gun)
4	Has been physically cruel to people
5	Has been physically cruel to animals
6	Has stolen while confronting a victim (for example, mugging, purse snatching, extortion or armed robbery)
7	Has forced someone into sexual activity
**Category 2: Destruction of property (DP)**	
8	Has deliberately engaged in fire setting with the intention of causing serious damage
9	Has deliberately destroyed others’ property (other than fire setting)
**Category 3: Deceitfulness or theft (DT)**	
10	Has broken into someone else’s house, building or car
11	Often lies to obtain goods or favours or to avoid obligations (e.g. ‘cons’)
12	Has stolen items of nontrivial value without confronting a victim (e.g. shoplifting, but without breaking and entering, or forgery).
**Category 4: Serious violation of rules (SVR)**	
13	Often stays out at night despite parental prohibitions, beginning before 13 years of age
14	Has run away from home overnight at least twice while living in the parental or parental surrogate home, or once without returning for a lengthy period
15	Is often truant from school, beginning before 13 years of age

A DSM5 diagnosis of CD is given when the following are met: (A) Repetitive and persistent pattern of behaviour in which the basic rights of others or major age-appropriate societal norms or rules are violated, as manifested by the presence of at least 3 of the above 15 criteria (20%) in the last 12 months from any of the 4 categories, with at least one criterion present in the last 6 months; (B) The disturbance in behaviour causes clinically significant impairment in social, academic or occupational functioning; (C) If the individual is 18 years of age or older, criteria are not met for antisocial personality disorder. Limited prosocial emotions specifier: This is applied to those who meet 20% criteria for CD and who also show two or more of the following symptoms over an extended period (that is 12 or more months) across multiple relationships and settings: (a) Lack of remorse or guilt, (b) Callous—lack of empathy, (c) A lack of concern about educational or occupational performance, (d) shallow emotions.

**Table 2 children-09-00963-t002:** Demographics (age, gender), PSC and CD scores of the total Uppsala Cross-sectional Life and Health Young Survey (LHS) study cohort (*n* = 3132).

(N), Mean % Score [s.d.]			
Variable	Males (*n* = 1460)	Females (*n* = 1672)	*p*-Value
Age (years) ^a^	16.4	16.44	0.293
	[1.15]	[1.14]	
Total PSC score (%) ^a^	42.08	48.76	<0.001
	[8.85]	[10.00]	
Psychological Complaint score (%) ^a^	39.43	46.94	<0.001
	[10.18]	[11.39]	
Somatic score (%) ^a^	44.09	50.14	<0.001
	[9.27]	[10.33]	
Total CD score (%) ^a^	33.29	32.04	<0.001
	[6.97]	[5.40]	
APA score (%) ^a^	31.29	30.44	<0.001
	[6.74]	[4.99]	
DT score (%) ^a^	26.23	24.94	<0.001
	[9.89]	[6.53]	
DP score (%) ^a^	28.99	26.11	<0.001
	[16.64]	[12.87]	
SVR (%) ^a^	45.21	43.67	<0.001
	[9.05]	[9.08]	

^a^*t*-test of mean difference; ^b^ Chi-Squared test of frequency distribution; BMI = Body Mass Index; PSC = Psychosomatic Complaints; CD = Conduct Disorder; APA=Aggression to People and Animals; DT = Deceit and Theft; DP = Destruction of Property; SVR=Serious Violation of Rules.

**Table 3 children-09-00963-t003:** ANOVA mean scores and standard deviations for each independent variable (Gender, Age, PSC Severity) on PSC and Total CD scores of the total Uppsala Cross-sectional Life and Health Young Survey (LHS) study cohort (*n* = 3132).

Variable	Gender Males (*n* = 1460) Females (*n* = 1672)	Gender *p*-Value	Age Younger (*n* = 1533) Older (*n* = 1599)	Age *p*-Value	PSC Severity Low (*n* = 1101) Mid (*n* = 913) High (*n* = 1118)	PSC Severity *p*-Value
Age (years)	16.4 [1.15] 16.4 [1.14]	0.293	15.4 [0.49] 17.4 [0.59]	<0.001	16.3 [1.13]	^a^ 0.047
					16.4 [1.12]	^b^ <0.001
					16.6 [1.16]	^c^ <0.001
Total PSC score (%)	51.8 [10.88] 60.0 [12.3]	<0.001	55.4 [12.40] 56.9 [12.28]	<0.001	35.6 [3.82]	^a^ <0.001
					44.7 [2.21]	^b^ <0.001
					56.7 [7.14]	^c^ <0.001
Psychological Complaint score (%)	20.9 [5.4] 24.9 [6.04]	<0.001	22.5 [6.06] 23.6 [6.05]	<0.001	33.7 [5.29]	^a^ <0.001
					42.9 [5.03]	^b^ <0.001
					55.8 [9.44]	^c^ <0.001
Somatic score (%)	30.9 [6.5] 35.1 [7.2]	<0.001	32.9 [7.21] 33.3 [7.21]	0.067	37.5 [4.74]	^a^ <0.001
					46.6 [4.10]	^b^ <0.001
					57.7 [7.67]	^c^ <0.001
Total CD score (%)	57.9 [12.13] 55.7 [9.4]	<0.001	56.0 [11.73] 57.5 [9.79]	<0.001	31.2 [5.45]	^a^ <0.001
					32.03 [5.0]	^b^ <0.001
					34.5 [7.24]	^c^ <0.001

**^a^** Low vs. Mid PSC Severity; ^b^ Mid vs. High PSC Severity; ^c^ Low vs. High PSC Severity.

**Table 4 children-09-00963-t004:** Mean scores and standard deviations for each independent variable (Gender, Age, PSC Severity) on the sub-scores of Conduct Disorder.

Variable	Gender Males (*n* = 1460) Females (*n* = 1672)	Gender *p*-Value	Age Younger (*n* = 1533) Older (*n* = 1599)	Age *p*-Value	PSC Severity Low (*n* = 1101) Mid (*n* = 913) High (*n* = 1118)	PSC Severity *p*-Value
Aggression to People and Animals Score (%)	27.0 [8.39] 25.0 [4.68]	<0.001	25.9 [7.47] 26.0 [5.97]	0.292	25.3 [6.43]	^a^ 0.177
					25.6 [5.35]	^b^ <0.001
					26.8 [7.89]	^c^ <0.001
Deceit and Theft Score (%)	26.2 [9.89] 24.9 [6.53]	<0.001	25.4 [8.96] 25.6 [7.59]	0.235	24.5 [7.89]	^a^ 0.048
					25.0 [6.85]	^b^ <0.001
					27.1 [9.45]	^c^ <0.001
Destruction of Property Score (%)	29.0 [16.6] 26.1 [12.9]	<0.001	27.7 [15.40] 27.2 [14.24]	0.192	25.9 [13.65]	^a^ 0.014
					27.3 [13.97]	^b^ 0.003
					29.1 [16.36]	^c^ <0.001
Serious Violation of Rules Score (%)	44.7 [9.61] 43.1 [9.59]	<0.001	41.7 [9.12] 45.9 [9.65]	<0.001	41.8 [8.44]	^a^ <0.001
					43.1 [9.23]	^b^ <0.001
					46.5 [10.41]	^c^ <0.001

^a^ Low vs. Mid PSC Severity; ^b^ Mid vs. High PSC Severity; ^c^ Low vs. High PSC Severity.

**Table 5 children-09-00963-t005:** Correlations between the subscales of Conduct Disorder Scores (%) in the Cross-sectional Life and Health Young Survey (LHS) study cohort (*n* = 3132).

Bivariate Correlation	Coefficient	*p* Value
APA × DT	0.753	<0.001
APA × DP	0.612	<0.001
DT × DP	0.625	<0.001
DT × SVR	0.384	<0.001
APA × SVR	0.363	<0.001
DP × SVR	0.293	<0.001

APA = Aggression to People and Animals; DT = Deceit and Theft; DP = Destruction of Property; SVR = Serious Violation of Rules.

## Data Availability

The datasets used and/or analysed during the current study are available from the corresponding author on request. SJB had full access to all data and takes responsibility for the integrity and accuracy of the data analysis.

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
