# Peer review of "Self-Reported Psychosomatic Complaints and Conduct Problems in Swedish Adolescents"

_children, 2022, doi:10.3390/children9070963_

Round 1
Reviewer 1 Report
Dear All,
A very important topic, which will contribute significantly to adolescents' health. However, I am here with some recommendations to improve the quality of work
I am not sure whether the researchers utilised recent or old data. If the researchers utilised old data back to 2011, they need to provide evidence that the situation did not change since then.
Please elaborate more on how the researchers maintained the ethical principles and protected the participants' identities.
I am not sure if the researchers who examined the data were able to identify the adolescents. If yes, what was the protocol that they followed in case the participants' records showed that they have conduct problems or/and psychosomatic complaints?
In table 2, researchers reported “height/weight for BMI”. This variable was never been introduced before. Researchers need to elaborate more about it in the methods section and what is the importance of this variable.
The researchers used ANOVA to test: “The effects of gender (boys, girls), age (younger, older adolescent) and PSC severity (low, medium, high) on total conduct problems score.”. However, researchers need to clearly mention that they used an independent t-test with the independent variables which consisted of two-level. Researchers mentioned this in table 2 but they failed to mention anything about the independent t-test in the plan for “Statistical analyses” and actual “Statistical analyses”. Please state clearly that the independent t-test was used with independent variables which consisted of two levels gender and adolescents’ age
In statistical analysis, I was expecting to see two tables one for the ANOVA and one for the MANOVA instead of the long two paragraphs, which are difficult to be followed. Please build two tables
The first table should show the difference in mean between the independent variables including age, gender etc, and the dependent continuous variables conduct problems, psychosomatic complaints, psychological complaints, and Somatic Complaints. The second is a MANOVA table.
The researchers did not use any correlational statistics like Pearson or Spearman to test the relationship between the dependent variables.
The discussion needs to be expanded and provide more information about the implication of the study.
Author Response
Comment 1. A very important topic, which will contribute significantly to adolescents' health. However, I am here with some recommendations to improve the quality of work
Authors’ response 1. We thank the reviewer for their positive comment.
Comment 2. I am not sure whether the researchers utilised recent or old data. If the researchers utilised old data back to 2011, they need to provide evidence that the situation did not change since then.
Authors’ response 2. Unfortunately we are not able to provide evidence that the situation has not changed since 2011, because Uppsala County Council conducted this specific data collection from 2005 until 2011. We have highlighted in the methods section and in the limitations section of the discussion that this is old data. However, we argue that the link between psychosomatic complaints and behavioural issues are less culturally bound across time than DSM diagnoses that do indeed change after each revision. On page 10, from line 410 we have added: “ In addition, this is now quite old data (from 2011), however, we argue that the link between psychosomatic complaints and problem behaviours in children are less culturally-bound than formal DSM-5 criteria, and so less susceptible to changes over time.”
Comment 3. Please elaborate more on how the researchers maintained the ethical principles and protected the participants' identities.
Authors’ response 3. On page 3, from line 127 we have added the following to clarify: “All respondents were asked to complete the survey once only, in one of these years, during school hours, and no identifiable personal data (e.g. date of birth, name, exact home location) was collected by the school researchers. Consent from parents and assent from pupils was collected, and both pupils and parents were made aware during in-troduction to the survey and the study itself that they could withdraw at any time, both participation and their data, without rights being affected.”
Comment 4. I am not sure if the researchers who examined the data were able to identify the adolescents. If yes, what was the protocol that they followed in case the participants' records showed that they have conduct problems or/and psychosomatic complaints?
Authors’ response 4. No, the researchers were not able to identify any of the adolescents recruited into the study. However, children were informed prior to the study that if they wanted to talk to someone at the school about any physical or mental health issues, they could do so confidentially.
Comment 5. In table 2, researchers reported “height/weight for BMI”. This variable was never been introduced before. Researchers need to elaborate more about it in the methods section and what is the importance of this variable.
Authors’ response 5. We apologise for this error – while height and weight were recorded, and we considered using BMI in our analyses, we did not do this in the current analyses. We have therefore removed BMI from the table.
Comment 6. The researchers used ANOVA to test: “The effects of gender (boys, girls), age (younger, older adolescent) and PSC severity (low, medium, high) on total conduct problems score.”. However, researchers need to clearly mention that they used an independent t-test with the independent variables which consisted of two-level. Researchers mentioned this in table 2 but they failed to mention anything about the independent t-test in the plan for “Statistical analyses” and actual “Statistical analyses”. Please state clearly that the independent t-test was used with independent variables which consisted of two levels gender and adolescents’ age
Authors’ response 6. On page 5, from line 221, we have now clarified the issue that we have used post-hoc t-tests to determine the direction and significance of any differences reported in the ANOVA/MANOVA analyses.
Comment 7. In statistical analysis, I was expecting to see two tables one for the ANOVA and one for the MANOVA instead of the long two paragraphs, which are difficult to be followed. Please build two tables
The first table should show the difference in mean between the independent variables including age, gender etc, and the dependent continuous variables conduct problems, psychosomatic complaints, psychological complaints, and Somatic Complaints. The second is a MANOVA table.
Authors’ response 7. We thank the reviewer for this recommendation. We have now updated the manuscript with the two tables as advised: one for ANOVA statistics and one for what would be the MANOVA analyses, however, in reference to comment 8 below – the sub scores of CD were highly correlated, and so we had to report these as separate ANOVAs.
Comment 8. The researchers did not use any correlational statistics like Pearson or Spearman to test the relationship between the dependent variables.
Authors’ response 8. We have now added a correlation table to demonstrate that the CD sub scale dependent variables are highly correlated. We did not do this for the first ANOVA as there was only one dependent variable (total conduct disorder score).
Comment 9. The discussion needs to be expanded and provide more information about the implication of the study.
Authors’ response 9. We have now substantially added some main implications of the study to the conclusions section on page 13, beginning at line 489.
Reviewer 2 Report
The study proposed for publication raises a very interesting question that relates psychosomatic complaints and their power in identifying the children and adolescents at risk of developing conduct disorders. The study is thorough and the sample is impressive in its size. The authors correctly identify the strengths and the weaknesses of the study while certainly bringing a clear contribution in identifying a consistent measure for potential behavioral disturbances a priori, based on things the target group is open to talk about.
However, there are some areas in which certain clarifications/completions could help:
Introduction
The causality effect relationship that seems to be the main issue of the paragraph requires further clarification. Are PSCs the result of emotional trauma or are they predicting a deviant behavior and emotional misregulation? I believe that stating clearly that they use bullying as a particular type of CD and use the data in the literature to make the argument could help.
Line 152. Missing dot after rating?
Results
In presenting the data I might suggest a correlation table. Is common for Anova analysis and easier to interpret.
Discussion
The first paragraph makes claims like: „especially those with higher PSC severity scores displaying higher levels of serious rule violations” or „older adolescents (17–18 years of age) reported higher levels of PSCs and conduct problems than younger adolescents” and it would be nice to see that in the results section (the table indicates a p<0.001 for all results, with no measure of differentiation between them).
Discussion/Conclusion
Although the case, under the circumstances of the study, is clearly made, I believe that a hint on how these kinds of results, i.e., early discovery of potential behavioral disturbances can help in preventing/mitigating them would be helpful in underlying the practical value of the study.
Author Response
Comment 1. The study proposed for publication raises a very interesting question that relates psychosomatic complaints and their power in identifying the children and adolescents at risk of developing conduct disorders. The study is thorough and the sample is impressive in its size. The authors correctly identify the strengths and the weaknesses of the study while certainly bringing a clear contribution in identifying a consistent measure for potential behavioral disturbances a priori, based on things the target group is open to talk about.
Authors’ response 1. We thank the reviewer for the positive feedback.
Comment 2. However, there are some areas in which certain clarifications/completions could help:
Authors’ response 2. We really appreciate the reviewer’s help to substantially improve our manuscript.
Introduction
Comment 3. The causality effect relationship that seems to be the main issue of the paragraph requires further clarification. Are PSCs the result of emotional trauma or are they predicting a deviant behavior and emotional misregulation? I believe that stating clearly that they use bullying as a particular type of CD and use the data in the literature to make the argument could help.
Authors’ response 3. We thank the reviewer for this excellent comment, and have now added to the introduction on page 2, line 80 the following sentence: “However, caution must be taken when considering the causal relationship between PSCs and deviant behavior such as bullying or other CD subtypes, because PSCs could also be the result of emotional trauma, affect dysregulation or some other cause (e.g. learned behaviour such as modelling).”
Comment 4. Line 152. Missing dot after rating?
Authors’ response 4. We thank the reviewer for their eagle eye! We have now added the missing dot after rating.
Results
Comment 5. In presenting the data I might suggest a correlation table. Is common for Anova analysis and easier to interpret.
Authors’ response 5. We thank the reviewer for this suggestion – the first reviewer also suggested this, and we have now added a correlation table as per your advice. We note that the sub scale scores for the CD variable are highly correlated, preventing us from running a MANOVA. We have now updated the manuscript to reflect the individual ANOVAs we had to run instead.
Discussion
Comment 6. The first paragraph makes claims like: „especially those with higher PSC severity scores displaying higher levels of serious rule violations” or „older adolescents (17–18 years of age) reported higher levels of PSCs and conduct problems than younger adolescents” and it would be nice to see that in the results section (the table indicates a p<0.001 for all results, with no measure of differentiation between them).
Authors’ response 6. We have now updated the tables (3 and 4) to reflect these differentiations – however, we do not illustrate trends in the table (e.g. the interactions in SVR result), only significant data. But trend data are available on request.
Discussion/Conclusion
Comment 7. Although the case, under the circumstances of the study, is clearly made, I believe that a hint on how these kinds of results, i.e., early discovery of potential behavioral disturbances can help in preventing/mitigating them would be helpful in underlying the practical value of the study.
Authors’ response 7. We thank the reviewer for this comment – a similar point was made by reviewer 1. We have now updated the conclusions section to better highlight the implications of this study.
Round 2
Reviewer 1 Report
Researchers fairly addressed all the suggested comments
Warm Regards